# `VIBE`: Annotation-Free Video-to-Text Information Bottleneck Evaluation for TL;DR

**Shenghui Chen,**[*] **Po-han Li,**[*] **Sandeep Chinchali,** **Ufuk Topcu**
The University of Texas at Austin
{shenghui.chen, pohanli, sandeepc, utopcu}@utexas.edu

## Abstract

Many decision-making tasks, where both accuracy and efficiency matter, still require human supervision. For example, tasks like traffic officers reviewing hour-long dashcam footage or researchers screening conference videos can benefit from concise summaries that reduce cognitive load and save time. Yet current vision-language models (VLMs) often produce verbose, redundant outputs that hinder task performance. Existing video caption evaluation depends on costly human annotations and overlooks the summaries' utility in downstream tasks. We address these gaps with **V**ideo-to-text **I**nformation **B**ottleneck **E**valuation (VIBE), an annotation-free method that scores VLM outputs using two metrics: *grounding* (how well the summary aligns with visual content) and *utility* (how informative it is for the task). VIBE selects from randomly sampled VLM outputs by ranking them according to the two scores to support effective human decision-making. Human studies on `LearningPaper24`, `SUTD-TrafficQA`, and `LongVideoBench` show that summaries selected by VIBE consistently improve performance—boosting task accuracy by up to $61.23\%$ and reducing response time by $75.77\%$ compared to naive VLM summaries or raw video. [2]

## 1 Introduction

Efficiently extracting relevant information from extensive video is a major bottleneck for human decision-making, where both accuracy and efficiency matter [1–4]. Tasks demanding human supervision, such as a traffic officer analyzing hours of dashcam footage to determine fault or a researcher distilling key insights from a lengthy oral presentation, are often limited by the time and cognitive load required to process raw video streams. In this work, we aim to improve the *quality* and *brevity* of video summaries to boost human task performance compared to existing vision-language model (VLM) outputs and raw video, especially for longer clips where summarization offers greater utility.

Existing video caption evaluation metrics, however, rely heavily on reference-based comparisons to human-annotated summaries [5–8]. These metrics face two main issues. First, these works require human annotators to watch video clips and write gold-standard captions, which contradicts the goal of reducing human response time and limits generalization to unseen video clips. Second, they are oblivious to downstream tasks and fail to measure how well captions support the tasks.

We propose **V**ideo-to-text **I**nformation **B**ottleneck **E**valuation (VIBE), an annotation-free method for selecting task-relevant video summaries without model retraining. As shown in Figure 1, VIBE defines two metrics—grounding and utility scores—based on the information bottleneck principle [9]. It uses pointwise mutual information to quantify how well a summary reflects video evidence and supports the downstream task. We leverage next-token prediction in VLMs to access the probability

---

[*]Equal contribution (Order determined by coin toss).

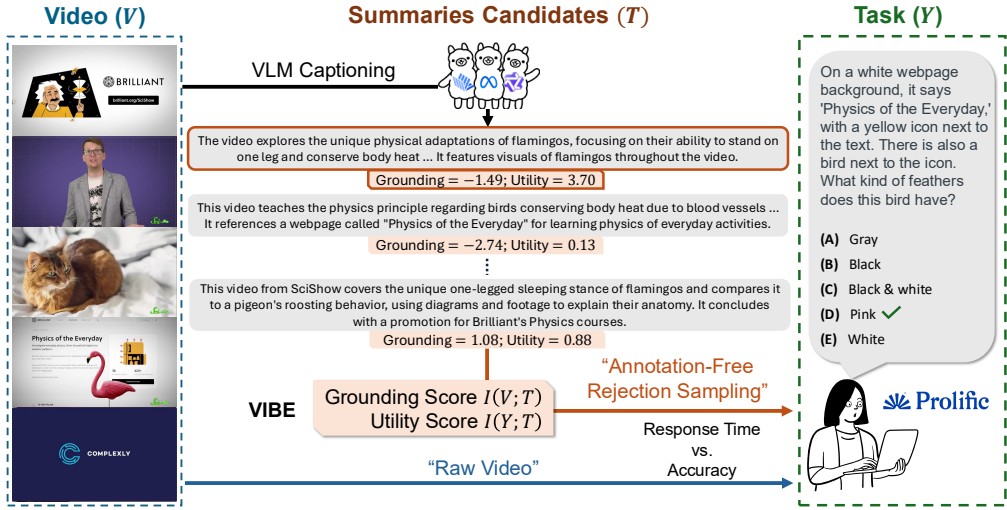

Figure 1: **VIBE for Video-to-Text Summary Selection.** Given a video, a task, and VLM-generated summaries, VIBE ranks the summaries using the proposed grounding and utility scores, which assess video alignment and task relevance. It selects the summary most conducive to helping human users achieve higher task accuracy and lower response time compared to watching the full video.

of generating summaries or task answers. By comparing these probabilities with and without key information, we measure how well one modality (text or video) compensates for missing information in the other to assess grounding and task relevance, as shown later in Figure 2. Using these scores, VIBE selects the most decision-supportive summary for humans from randomly sampled VLM outputs via annotation-free rejection sampling, which filters candidates without human labels for faster task completion than watching full video clips.

To evaluate VIBE, we conduct between-subjects user studies with 243 participants across three datasets—LearningPaper24 (self-curated), LongVideoBench [10], and SUTD-TrafficQA [11]—measuring human performance in terms of accuracy, response time, and inverse efficiency score, the ratio of response time to accuracy [12]. Results show that summaries selected with maximal utility score improve task accuracy by up to 40%, while those chosen by maximizing grounding score yield up to 27.6% gains, both on the LongVideoBench dataset. VIBE-selected summaries also significantly reduce response time compared to raw video. We also observe a strong positive correlation between utility score and human accuracy, and a strong negative correlation between summary length and response time per word. These patterns highlight the value of concise, relevant information for efficient human decision-making.

**Contributions.**   Our contributions are threefold: (a) We identify the need and propose the problem of annotation-free, task-aware evaluation for video-to-text summarization, improving human response time and accuracy without relying on gold-standard captions. (b) We propose VIBE, an annotation-free evaluation framework that combines grounding and utility scores of video clips to rank and select high-quality summaries from VLM outputs without requiring retraining. (c) We demonstrate through user studies that VIBE summaries significantly boost humans' task accuracy by up to 61.23% and reduce response time by up to 75.77% compared to standard VLM outputs.

**Critique and Open Problems.**   This work reframes the video-to-text evaluation problem through the lens of human decision support, offering an annotation-free, scalable alternative to costly reference-based comparisons. VIBE's ability to score and select summaries without training makes it a practical plug-in for both closed- and open-source VLMs. Looking ahead, VIBE opens several promising directions. One is exploring the joint optimization of summary generation and selection through self-supervised fine-tuning of VLMs. Another is extending beyond human supervision tasks to investigate whether task-aware captions can improve VLM performance on downstream reasoning tasks. These directions point to the broader applicability of VIBE beyond evaluating video summarization.

## 2 Related Work

**Reference-based Video Caption Evaluation.** Evaluating video caption quality is crucial for tasks like video question answering [13, 14], text-to-video retrieval [5, 15, 16], and multimodal language model training. The quality of captions greatly impacts the downstream performance of tasks and models. While existing benchmarks compare VLM-generated captions to gold-standard references using metrics like ROUGE [17], BLEU [18], and CIDEr [19], these are costly to curate and focus solely on captions rather than video context. In contrast, we propose VIBE, a new evaluation metric that incorporates video context, requires no gold standard, and can be applied to unseen video-caption pairs. VIBE extends the information-theoretic approach of [20], which uses pointwise mutual information to assess news summaries for language model tuning. We adapt it for video captions and examine their effect on human task accuracy and response time.

**Human-Centric Evaluation via Response Time and Accuracy.** In the context of human-agent interaction involving natural language [21], traditional human evaluations have focused on assessing summaries based on fluency and informativeness [22, 23], but these methods are often subjective and hard to scale. To address this, we adopt an extrinsic evaluation approach that measures how summaries affect human performance on downstream tasks [24, 25]. Specifically, we evaluate captions based on human response time and accuracy on multiple-choice questions grounded in video content. To account for the speed-accuracy tradeoff [26], we also report the inverse efficiency score [12], which normalizes response time by accuracy.

## 3 Preliminaries

**Mutual Information.** Our work heavily relies on the concept of mutual information. For the ease of readers, we briefly introduce it here. Mathematically, for two random variables, $X, Z$, we can calculate their mutual information:

$$\mathbf{I}(X; Z) = \mathbf{E}\left[\log \frac{\mathbf{P}(X, Z)}{\mathbf{P}(X)\mathbf{P}(Z)}\right] = \mathbf{E}\left[\log \frac{\mathbf{P}(X|Z)}{\mathbf{P}(X)}\right], \quad \text{(Mutual Information)} \quad (1)$$

where the expectation operator is over the joint probability distribution of $X$ and $Z$.

Intuitively, mutual information measures the reduction in uncertainty about $X$ after observing $Z$. It is zero if and only if $X$ and $Z$ are independent, meaning knowledge of $Z$ provides no information about $X$. Higher mutual information indicates stronger dependency between the two variables. In this work, we leverage mutual information to quantify how much the summaries retain relevant information from the raw video clips and how well they support the target downstream task.

**Information Bottleneck.** The information bottleneck (IB) framework extracts relevant information from input data while compressing irrelevant details. Tishby et al. [9] first introduced it to formalize the trade-off between accuracy and complexity in learning systems. Later, researchers applied it to domains such as neuroscience [27], natural language processing [28, 29], and computer vision [30, 31]. To ensure clarity, we modify the notation from [9] to align with our later sections, where $V$ is the raw video input, $T$ is the summary of the video, and $Y$ is the downstream task of our interest.

We now introduce the IB framework, which seeks to learn a representation $T$ that discards sensitive and irrelevant information from the input $V$ while preserving information useful for predicting the target task $Y$. IB relies on minimizing the mutual information between input $V$ and a compressed representation $T$, while preserving as much information as possible about a target variable $Y$:

$$\min_{\mathbf{P}(t|v)} \underbrace{\mathbf{I}(V; T)}_{\text{compression}} - \beta \underbrace{\mathbf{I}(T; Y)}_{\text{informativeness}}, \quad \text{(Information Bottleneck)} \quad (2)$$

where $\mathbf{I}(\cdot\,;\cdot)$ denotes mutual information, and $\beta \geq 0$ controls the trade-off between compression and prediction. The optimization variable is the conditional distribution $\mathbf{P}(t|v)$, which defines a stochastic encoder that maps $v \in V$ to $t \in T$. The IB principle in eq. (2) formalizes the trade-off between compression and prediction. Minimizing $\mathbf{I}(V; T)$ encourages stronger compression, while maximizing $\mathbf{I}(T; Y)$ promotes the preservation of task-relevant information. The parameter $\beta$ balances these two competing objectives: a higher $\beta$ prioritizes retaining more information about $Y$, while a lower $\beta$ encourages stronger compression of $V$.

Researchers use the IB principle to analyze generalization, regularization, and the role of hidden representations of neural networks [32, 33], where $V$ is the input of neural networks, $T$ is the output of intermediate layers, and $Y$ is the classification label. It also drives recent advances in representation learning, where models aim for compact, informative encodings [34, 35].

# 4 Video-to-Text Information Bottleneck Evaluation (VIBE)

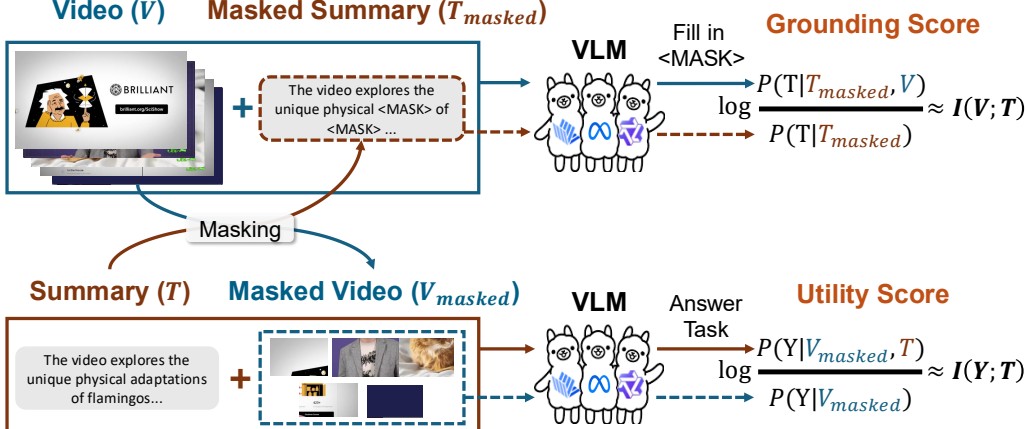

Figure 2: **Computing VIBE Scores via Masked Inference.** VIBE estimates grounding and utility scores using the next-token prediction mechanism of VLMs. The grounding score measures how well the video helps reconstruct a masked summary, while the utility score captures how much the summary improves task prediction given a masked video.

We now formally describe our video-to-text information bottleneck evaluation (VIBE) method. Inspired by the IB formulation in eq. (2), we adapt its two terms for the purpose of evaluating video-to-text summary quality. We denote the raw video clips that are fed into the VLMs as $V$, the summaries from the VLM response as $T$, and the task target we care about as $Y$. For instance, let $V$ be a video of a paper presentation, $T$ be the summary of the presentation, and the task $Y$ be to determine the primary area to see if it is of one's interest.

**Grounding Score.** In eq. (2), the mutual information $\mathbf{I}(V; T)$ between raw data and representation is the compression term. In our setting, we reinterpret it as the *grounding score*, which quantifies how grounded the textual summary is to the video. The higher this score is, the more closely the summary is anchored to the video.

However, one cannot access the joint and marginal probability distributions over the occurrences of video clips or text summaries. Therefore, we can only calculate the empirical mutual information $\log \frac{\mathbf{P}(X,Z)}{\mathbf{P}(X)\mathbf{P}(Z)}$ without the expectation operator to approximate the $\mathbf{I}(V; T)$ term. Literature refers to the empirical term without the expectation as pointwise mutual information [36], which we denote as $\mathbf{I}^P(\cdot; \cdot)$. Unlike mutual information, which must be non-negative as it is the expectation of pointwise mutual information, pointwise mutual information ranges from $[-\infty, \infty]$.

We now explain the approximation technique in VIBE to calculate the grounding score. Recall from eq. (1), we rewrite pointwise mutual information as:

$$\mathbf{I}^P(V; T) = \log \frac{\mathbf{P}(T|V)}{\mathbf{P}(T)} \approx \log \frac{\mathbf{P}(T|V, T_{\text{masked}})}{\mathbf{P}(T|T_{\text{masked}})}. \quad \text{(Grounding Score)} \quad (3)$$

We condition both the numerator and denominator with $T_{\text{masked}}$, a masked version of the text, to approximate $\mathbf{I}^P(V; T)$. The approximation holds if $T_{\text{masked}}$ is independent of both the video $V$ and the original text $T$. Masking key content makes $T_{\text{masked}}$ effectively uninformative and thus independent. We put the detailed derivation in Appendix I. Following Jung et al. [20], we use tf-idf [37] to identify keywords and replace them with a "<MASK>" token. This masking helps ensure that eq. (3) reflects the true dependency between $V$ and $T$, rather than exploiting shortcuts from the text

itself. Thus, the grounding score measures how much the TL;DR remains genuinely grounded in the video beyond what the masked text alone would suggest.

To compute eq. (3), we run two separate inferences with the same VLM to define the ratio of two conditional probabilities $\mathbf{P}(\cdot|\cdot)$. The first, $\mathbf{P}(T|V, T_{\text{masked}})$, is the product of next-token prediction probabilities when reconstructing the masked text given both the video and the masked input. The second, $\mathbf{P}(T|T_{\text{masked}})$, is computed similarly but without providing the video. Only by conditioning on the masked input can we leverage the VLM's next-token prediction probabilities to estimate these likelihoods. By dividing these two conditional probabilities and taking the logarithm, we approximate the pointwise mutual information, allowing us to estimate the grounding score.

**Utility Score.** The second term in eq. (2), the informativeness term, is the mutual information $\mathbf{I}(T; Y)$ between the representation and the target task. We refer to it as the *utility score*, which measures how useful the summary is for solving the downstream task. A higher utility score indicates that the summary preserves more information relevant to predicting or completing the target task. Similar to the grounding score, directly computing $\mathbf{I}(T; Y)$ is intractable because we cannot access the true probability distributions. Again, we approximate it using the pointwise mutual information with conditional probabilities:

$$\mathbf{I}^P(T; Y) = \log \frac{\mathbf{P}(Y|T)}{\mathbf{P}(Y)} \approx \log \frac{\mathbf{P}(Y|T, V_{\text{masked}})}{\mathbf{P}(Y|V_{\text{masked}})}. \text{ (Utility Score)} \quad (4)$$

Similar to the grounding score in eq. (3), $V_{\text{masked}}$ denotes video clips with key information removed to reduce their direct informativeness about both the summary $T$ and the task label $Y$. To compute the utility score in eq. (4), we compare two model inferences: one conditioned on both the summary $T$ and the masked video $V_{\text{masked}}$, and the other on $V_{\text{masked}}$ alone. Since both use the same masked visual input, any improvement in predicting $Y$ must come from the information provided by $T$. Intuitively, if a summary is useful, it should compensate for the missing content in the masked video and help recover task-relevant information. Thus, the utility score quantifies how well the summary restores information necessary for predicting the task, directly measuring the summary's contribution to downstream decision-making.

**Rejection Sampling over Grounding and Utility Scores.** We now describe how VIBE optimizes grounding and utility scores to select more informative and task-relevant video-to-text summaries for humans, as shown in Figure 1. Given multiple summaries $T$ sampled from the VLM's output distribution $P(T|V)$, VIBE selects the candidate that maximizes a weighted sum of the two pointwise mutual information terms:

$$\arg\max_{T \sim P(T|V)} \alpha \, \mathbf{I}^P(V; T) + \beta \, \mathbf{I}^P(T; Y). \text{ (TL;DR Selection with VIBE)} \quad (5)$$

Here, $\alpha, \beta$ are hyperparameters controlling the trade-off between grounding and utility in eq. (5). A higher $\alpha$ emphasizes alignment with video, favoring summaries that faithfully reflect the video, while a higher $\beta$ prioritizes task relevance, selecting summaries that improve downstream performance.

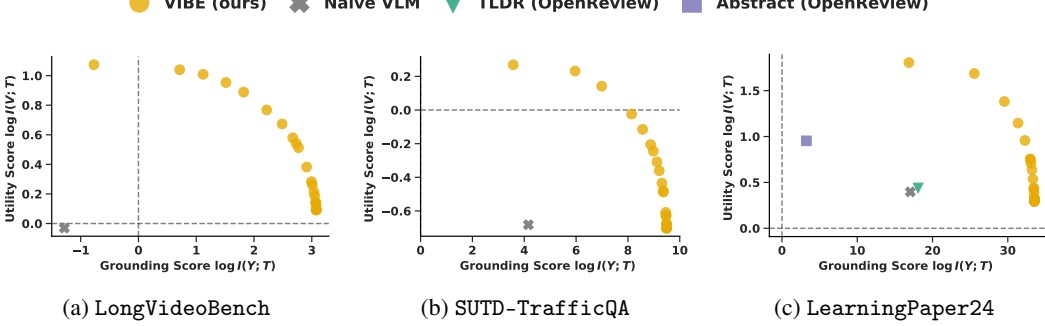

(a) `LongVideoBench`  (b) `SUTD-TrafficQA`  (c) `LearningPaper24`

Figure 3: **Pareto front of VIBE.** Summaries selected by VIBE form a Pareto frontier across different $(\alpha, \beta)$ per Equation (5), demonstrating optimal trade-offs between grounding and utility. In contrast, Naive VLM summaries, author-written TLDRs, and abstracts from OpenReview in `LearningPaper24` fall inside the frontier, indicating suboptimal scores.

Our formulation differs from traditional IB, where compression serves as a regularizer during model training to mitigate overfitting. In contrast, VIBE does not train a model—it selects a summary. Overfitting is therefore not a concern, and maximizing both mutual information terms is desirable to produce summaries that are both useful and faithful. To this end, we consider $\alpha, \beta \geq 0$ to jointly maximize both scores. Then, to explore potential trade-offs between the two scores, we perform a convex combination sweep over $\alpha \in \{0, 0.05, 0.1, \ldots, 1.0\}$, with $\beta = 1 - \alpha$. Each $(\alpha, \beta)$ pair is applied consistently across all datasets. This linear scalarization technique, commonly used in multi-objective optimization [38], produces the convex ("Pareto curve") shown in Figure 3, empirically revealing the inherent trade-off between grounding and utility scores.

Our results based on grounding and utility scores across sampled summaries from multiple datasets reveal a consistent trade-off between the two objectives. As shown in Figure 3, summaries selected by VIBE with varying $\alpha$ and $\beta$ form a Pareto front, capturing the inherent trade-off between the two VIBE scores. In contrast, randomly sampled summaries (*Naive VLM*) consistently fall inside this Pareto curve across datasets, indicating suboptimal grounding and utility. We observe this pattern when evaluating the two VIBE scores of VLM-generated summaries on tasks from `LearningPaper24`, `SUTD-TrafficQA`, and `LongVideoBench`. Summaries with high grounding scores tend to mirror the video content closely but often include redundant or irrelevant details for the task. In contrast, summaries with high utility scores boost task accuracy and response efficiency but may overlook important visual context. This trade-off reflects the core tension in the IB framework. By tuning $\alpha$ and $\beta$, VIBE adjusts this balance to match the needs of different downstream applications. We detail the experimental settings in the next section.

As shown empirically in the experiments, higher utility scores correlate with higher human task accuracy. Thus, one can use VIBE to select summaries with higher utility scores and present only the most informative summaries to humans, filtering out less useful ones. Notably, calculating the utility score requires access to task labels $Y$, but not gold-standard, human-annotated labels for summary $T$ as in previous works. When task labels are unavailable, VIBE can select summaries based on the highest grounding score, an unsupervised measure of alignment between the video and the generated summaries. Although the grounding score alone yields lower human performance than the utility score, our user study shows it still improves task accuracy compared to unfiltered VLM summaries.

**From Evaluation to Real-World Use.** VIBE evaluates the quality of summaries only from the VLM perspective, but its connection to human decision-making performance remains unanswered solely by the formulation. To bridge this gap, we conduct user studies measuring how different summary qualities, as evaluated by VIBE, impact human decision-making performance in Section 5. VIBE relies solely on text generation and next-token probability access, using black-box access to VLMs like the OpenAI API [39]. It enables VIBE to work with both closed- and open-source VLMs and scale well to real-world scenarios.

## 5 Experiment

We conduct user studies across three diverse datasets to evaluate how VIBE-generated summaries support human decision-making. The study measures participants' accuracy and response time in answering questions about video content. Here, we use Qwen2.5-VL-72B-AWQ [40] to evaluate VIBE for all datasets. A representative qualitative example from the study is provided in Appendix A. We also show that VIBE works across various VLMs through ablation studies.

### 5.1 Datasets

For evaluation, we select three datasets varying in domain, duration, and task nature.

**LearningPaper24** We introduce `LearningPaper24`, a curated dataset of 2,287 video presentations from ICLR 2024 and NEURIPS 2024, filtered based on key criteria: a valid OpenReview ID, an accessible SlidesLive video, an author-provided TL;DR and abstract, and a clearly defined primary area. Details on the curation process and dataset statistics are provided in Appendix C. The task associated with this dataset is to identify the primary area of each paper from 12 options. See instruction details in Appendix E.

**LongVideoBench & SUTD-TrafficQA**  `LongVideoBench` [10] features long instructional clips with QA pairs for extended reasoning, while `SUTD-TrafficQA` [11] contains short traffic clips with multiple-choice questions on causal and temporal understanding. Participants answer one question per each `LongVideoBench` clip; four questions per each `SUTD-TrafficQA` clip.

**Preprocessing**  In all datasets, we mask text by removing keywords with high tf-idf scores. For `LearningPaper24`, which features slide videos, we extract and mask keywords in the slide using EasyOCR [41]. For the other two datasets, we apply random $1/16$ cropping to all video frames. We use a subset of each dataset for VIBE evaluation and user studies (details provided in Appendix D).

## 5.2   User Study Design

For each dataset, 10 video stimuli with multiple-choice questions are shown in randomized order. We adopt a *between-subjects design* where participants are assigned to one of the four conditions listed below, and the format in which the stimulus is presented varies by the condition.

**Independent Variables.**   Participants are assigned to one of four conditions. In the **Video Only** condition, they watch the original video without text. In the remaining conditions, they view only a VLM-generated summary: a randomly selected VLM summary (**Naive**), the top-ranked summary from $k$ response candidates[3] by utility score (**Max-U**) or by grounding score (**Max-G**).

**Dependent Measures.**   We report three metrics: **accuracy**, the proportion of correct responses across 10 stimuli; **response time**, the time (in seconds) spent reading or watching each stimulus and answering its corresponding questions; and **inverse efficiency score (IES)**, the ratio of response time to accuracy, to account for speed–accuracy trade-offs [12].

**Hypotheses.**   We evaluate the following hypotheses: (**H1**) Participants in the Max-U and Max-G will achieve higher accuracy than those in Naive VLM. (**H2**) Participants in the Max-U and Max-G will respond more quickly than those in the Video Only. (**H3**) Max-U and Max-G will yield lower (i.e., more efficient) IES scores than Video Only, reflecting a better speed–accuracy trade-off.

**Participants.**   We recruit 243 participants across three datasets: 92 for `LearningPaper24`, 82 for `SUTD-TrafficQA`, and 69 for `LongVideoBench`. For `LearningPaper24`, participants are primarily CS graduate students or prescreened degree holders on Prolific [42]; participants for the other datasets are recruited generally. The average age is $37.59 \pm 11.06$ years, with a gender distribution of $63.37\%$ male, $35.80\%$ female, and $0.82\%$ non-binary. Further details are provided in Appendix F.

## 5.3   User Study Results and Analysis

We assess statistical significance using the independent t-test, reporting the t-statistic $t(df)$ (with degrees of freedom $df$), the significance level $p$, and the effect size measured by Cohen's $d$.

**On H1 (Accuracy).**   As shown in Table 1 and (a1, b1) of Figure 4 (with Max-U in yellow and Max-G in purple above Naive VLM in pink), both Max-U and Max-G consistently outperform the Naive VLM across all datasets, confirming **H1**. Max-U achieves the largest gains, with statistically significant improvements over Naive VLM in `LearningPaper24` ($t(36) = 2.486, p = 0.009, d = 0.806$), `LongVideoBench` ($t(26) = 3.215, p = 0.002, d = 1.261$), and `SUTD-TrafficQA` ($t(25) = 3.311, p = 0.001, d = 1.325$), likely due to its focus on task-relevant content that helps users quickly locate key information. However, computing the utility score requires task labels, which may limit its general applicability and scalability. Max-G, while yielding smaller gains, still significantly outperforms the baseline (e.g., `LearningPaper24`: $t(33) = 1.750, p = 0.045, d = 0.594$; `LongVideoBench`: $t(25) = 2.691, p = 0.006, d = 1.077$; `SUTD-TrafficQA`: $t(28) = 1.759, p = 0.045, d = 0.667$). Crucially, calculating the grounding score is fully self-supervised and does not require task labels, making it an alternative that still delivers meaningful accuracy gains.

---

[3]$k = 5$ in this study, and the $k$ responses are generated with various temperatures.

| Dataset | Duration (s) | Metric | IV Conditions | | | |
|---|---|---|---|---|---|---|
| | | | *Video Only* | *Naive* | *Max-G (ours)* | *Max-U (ours)* |
| LearningPaper24 | 250–325 | Acc ↑ | $23.50 \pm 10.62$ | $28.42 \pm 12.68$ | $35.00 \pm 7.91$ | $\mathbf{37.89 \pm 10.04}$ |
| | | RT ↓ | $192.73 \pm 108.01$ | $47.48 \pm 20.31$ | $61.13 \pm 39.33$ | $\mathbf{46.69 \pm 25.06}$ |
| | | IES ↓ | $9.85 \pm 6.63$ | $1.81 \pm 1.51$ | $1.85 \pm 1.20$ | $\mathbf{1.28 \pm 0.66}$ |
| LongVideoBench | 221–489 | Acc ↑ | $\mathbf{74.44 \pm 14.23}$ | $46.43 \pm 13.94$ | $59.23 \pm 9.17$ | $65.00 \pm 15.47$ |
| | | RT ↓ | $202.35 \pm 87.26$ | $86.50 \pm 63.33$ | $71.93 \pm 35.96$ | $\mathbf{65.86 \pm 41.34}$ |
| | | IES ↓ | $2.93 \pm 1.48$ | $1.83 \pm 1.01$ | $1.25 \pm 0.67$ | $\mathbf{1.14 \pm 0.80}$ |
| SUTD-TrafficQA | 2–10 | Acc ↑ | $82.86 \pm 5.08$ | $76.96 \pm 6.89$ | $80.47 \pm 3.21$ | $\mathbf{84.81 \pm 4.65}$ |
| | | RT ↓ | $\mathbf{48.63 \pm 21.30}$ | $79.46 \pm 25.77$ | $80.97 \pm 41.11$ | $137.27 \pm 81.64$ |
| | | IES ↓ | $\mathbf{0.59 \pm 0.27}$ | $1.05 \pm 0.37$ | $1.01 \pm 0.52$ | $1.66 \pm 1.02$ |

Table 1: Human performance (mean ± standard deviation) across three datasets under different IV conditions (detailed in Section 5.2). Metrics: Accuracy (Acc, %, higher is better), Response Time (RT, seconds, lower is better), and Inverse Efficiency Score (IES = RT/Acc, lower is better). Bolded values indicate the best performance among the IV conditions for each metric.

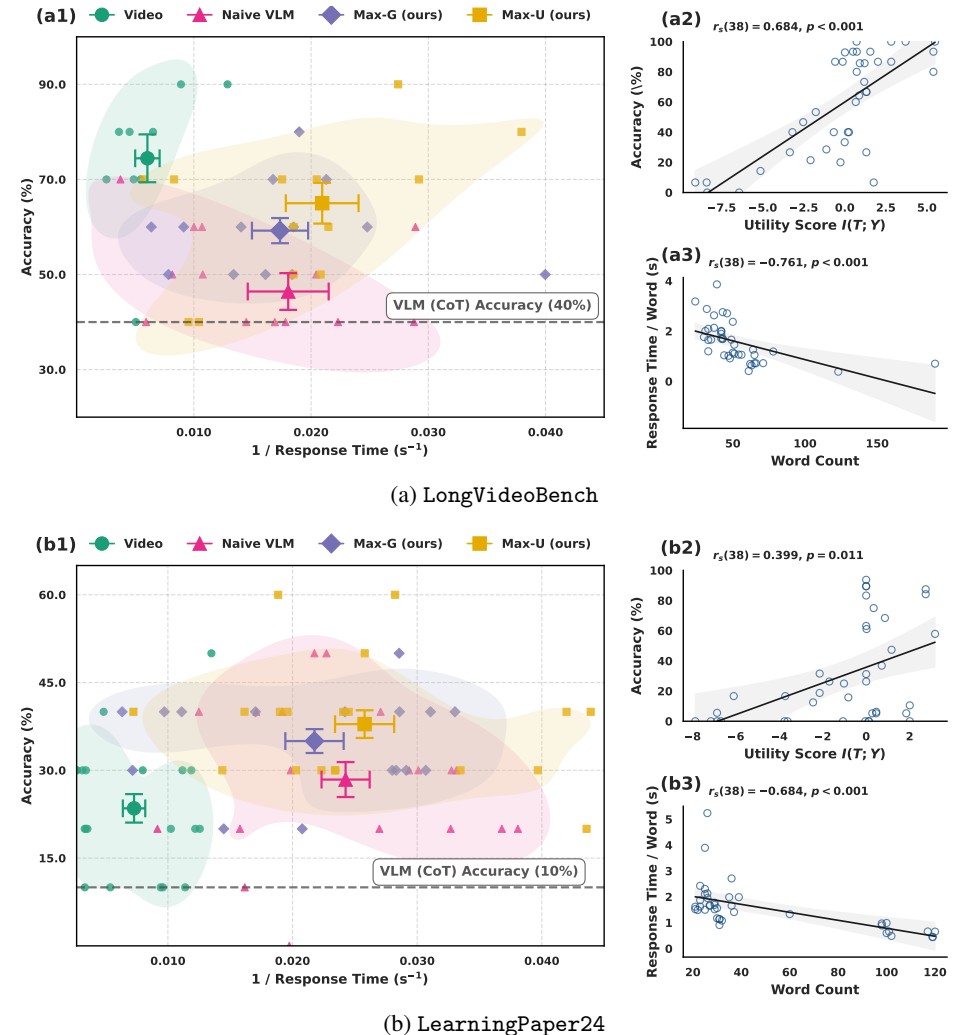

Figure 4: **(a1, b1)** Accuracy versus inverse response time. Each point represents an individual participant; large markers indicate group means with standard error of the mean. Shaded areas denote 2D kernel density estimates (threshold = 0.45). **(a2, b2)** Correlation between accuracy and utility score. **(a3, b3)** Correlation between response time per word and word count.

**On H2 (Response Time).** Table 1 shows that both Max-U and Max-G significantly reduce response times compared to the Video baseline in `LearningPaper24` ($t(37) = -5.599, p < .001, d = -1.794$) and `LongVideoBench` ($t(34) = -4.503, p < .001, d = -1.510$). However, this trend does not extend to `SUTD-TrafficQA`, where response time shows no consistent improvement. It is not surprising, as the significantly shorter video durations (2–10 seconds, compared to over 3 minutes in the other datasets) inherently limit the benefits of summarization. These results align with the expectation that VIBE is most effective in reducing cognitive load for long video clips.

**On H3 (IES).** As shown in Table 1, Max-U significantly outperform the Video Only baseline in `LearningPaper24` ($t(37) = -5.458, p < .001, d = -1.749$) and `LongVideoBench` ($t(21) = -3.592, p = .001, d = -1.617$), with Max-G showing similar gains (`LearningPaper24`: $t(34) = -4.631, p < .001, d = -1.553$; `LongVideoBench`: $t(20) = -3.421, p = .001, d = -1.567$). Furthermore, Max-U also outperform Naive VLM ($t(26) = -1.940, p = .032, d = -0.761$) in `LongVideoBench`. These results highlight VIBE's effectiveness in settings where processing full video clips is costly. In contrast, `SUTD-TrafficQA` shows limited IES improvement: despite Max-U's accuracy improvements over Naive VLM and Video Only, the brevity of the clips and the visual nature of fine-grained actions reduce the benefit of text-based summaries.

**Correlation Analyses.** We explore the correlation between utility score, grounding score, word count, and human performance (accuracy and response time) using Spearman's rank coefficient [43], which is robust to outliers and non-linear relationships. Full plots are provided in Appendix G. Two consistent trends hold across datasets: First, utility score positively correlates with accuracy: strongly in `LongVideoBench` ($r_s = 0.684, p < .001$), and moderately in `LearningPaper24` ($r_s = 0.399, p = .011$) and `SUTD-TrafficQA` ($r_s = 0.463, p = .004$). Second, word count is strongly negatively correlated with response time per word (`LongVideoBench`: $r_s = -0.761$, `LearningPaper24`: $r_s = -0.684$, `SUTD-TrafficQA`: $r_s = -0.658$, all $p < .001$), implying that longer summaries lead to faster processing per word, possibly due to reduced engagement and more shallow reading.

**Key Takeaways.** Max-U and Max-G outperform Naive VLMs in accuracy, response time, and inverse efficiency score, with Max-U providing the most significant improvements, especially for longer video clips. Max-G, though offering smaller gains, remains a self-supervised alternative that improves accuracy without task labels. The benefits of summarization are less pronounced in shorter video clips, like those in `SUTD-TrafficQA`, where brief durations limit gains. Correlation analyses show the VIBE utility scores align with higher accuracy, while longer summaries tend to increase unit response time, highlighting the importance of concise, relevant information.

## 5.4 Ablation over VLM Variants

VIBE scores can inherit bias from the underlying VLMs due to their training data and conditional probability estimates used in mutual information computation. To assess robustness under such model-induced bias, we compare VLMs of varying size and source. As shown in Appendix B, different backbones produce consistent IB score trends and scales. This consistency suggests that lightweight models can efficiently verify outputs from larger ones. To further reduce bias, one can evaluate a single summary using multiple VLMs—an ensemble-style strategy aligned with mixture-of-experts [44, 45] and model selection techniques [46]. This reflects the intuition that good outputs are often hard to generate, but easy to verify.

**Limitations.** Despite being training-free, VIBE requires multiple VLM inferences for grounding score evaluation, with the number of masked tokens influencing inference steps. Its pointwise mutual information estimate may be biased by two main factors: (a) the text and video masking strategy, and (b) the conditional probability modeling of the VLMs. For (a), the tf-idf score we use to select keywords for text masking requires hyperparameter tuning, and the optimal masking strategy remains an open question. For (b), VLMs introduce bias through their training data and modeling assumptions in computing eq. (5). Using separate models for generation and evaluation, as discussed in Section 5.4, can help mitigate this bias.

# 6 Conclusion

In this work, we introduce VIBE, an annotation-free framework for evaluating and selecting video-to-text summaries to support human decision-making. Unlike traditional caption evaluation metrics that rely on human-written references, VIBE uses information-theoretic scores—utility and grounding—to assess how well a summary supports a downstream task and aligns with video evidence. This enables scalable, task-aware summary selection without the need for retraining or annotations. Through a large-scale user study spanning three diverse datasets, we show that summaries selected by VIBE significantly improve both task accuracy and response time, especially for longer video clips where information overload is more likely.

## Acknowledgement

This work was supported in part by the National Science Foundation grants No. CNS-1836900, 2148186, the Office of Naval Research (ONR) under Grant No. N00014-22-1-2254, N00014-24-1-2097, the Defense Advanced Research Projects Agency (DARPA) contract FA8750-23-C-1018, and DARPA ANSR: RTXCW2231110. Any opinions, findings, and conclusions or recommendations expressed in this material are those of the authors and do not necessarily reflect the views of the National Science Foundation.

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

# Appendix

## A  Qualitative Result

To illustrate differences across summary generation methods, we present a qualitative example. In Table 3, each condition—Naive, Max-U, Max-G, and CoT—produces a distinct response based on the same video input (keyframe thumbnails shown in Figure 5). For better comparison, we also provide the original TL;DR and abstract from the authors on OpenReview in Table 2. We highlight key terms derived from OpenReview metadata and tf-idf analysis by masking them in gray.

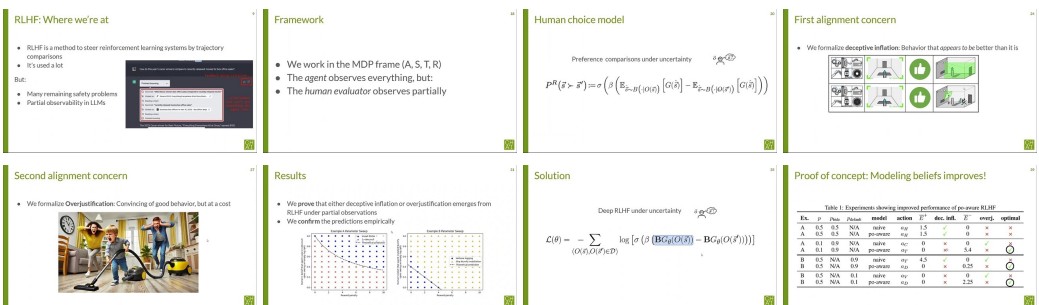

Figure 5: Keyframe thumbnails from the qualitative example SlidesLive talk. The correct answer is *(I) Human-AI Interaction and Ethics (Privacy, Fairness & Safety)*, while the VLM prediction is *(F) Reinforcement Learning & Robotics*.

| Type | Paper Summary |
|---|---|
| **TL;DR** | We study the challenges that arise when learning reward functions with human feedback from partial observations |
| **Abstract** | Past analyses of reinforcement learning from human feedback ( RLHF ) assume that the human evaluators fully observe the environment. What happens when human feedback is based only on partial observations? We formally define two failure cases: deceptive inflation and overjustification. Modeling the human as Boltzmann-rational w.r.t. a belief over trajectories, we prove conditions under which RLHF is guaranteed to result in policies that deceptively inflate their performance, overjustify their behavior to make an impression, or both. Under the new assumption that the human's partial observability is known and accounted for, we then analyze how much information the feedback process provides about the return function. We show that sometimes, the human's feedback determines the return function uniquely up to an additive constant, but in other realistic cases, there is irreducible ambiguity. We propose exploratory research directions to help tackle these challenges and experimentally validate both the theoretical concerns and potential mitigations, and caution against blindly applying RLHF in partially observable settings . |

Table 2: Author-provided TL;DR and abstract on OpenReview.

| Condition | Summary Response |
|-----------|------------------|
| **Naive** | This paper explores the challenges of partial observability in reinforcement learning from human feedback, focusing on deceptive inflation and overjustification in AI systems . |
| **Max-U** | This research investigates challenges in ensuring AI safety and addressing partial observability through advanced frameworks in human - machine interactions and reinforcement learning. |
| **Max-G** | The presentation discusses the challenges of partial observability in reinforcement learning from human feedback, focusing on deceptive inflation and overjustification in AI systems . |
| **CoT** | The slides discuss challenges in reinforcement learning from human feedback , particularly focusing on partial observability and the issues of deceptive inflation and overjustification. The content delves into the framework of Markov Decision Processes (MDPs) and the human choice model, which are central to reinforcement learning. The presentation also addresses safety problems and alignment concerns, which are critical in the context of human-AI interaction. Given the focus on reinforcement learning and the interaction between AI and human evaluators, the most suitable area for this paper is one that encompasses both reinforcement learning and the ethical considerations of human- AI interaction. |

Table 3: Responses under each condition for the qualitative example, keywords from OpenReview and tf-idf masked in grey.

## B    Ablation Study on Evaluating with Various VLMs

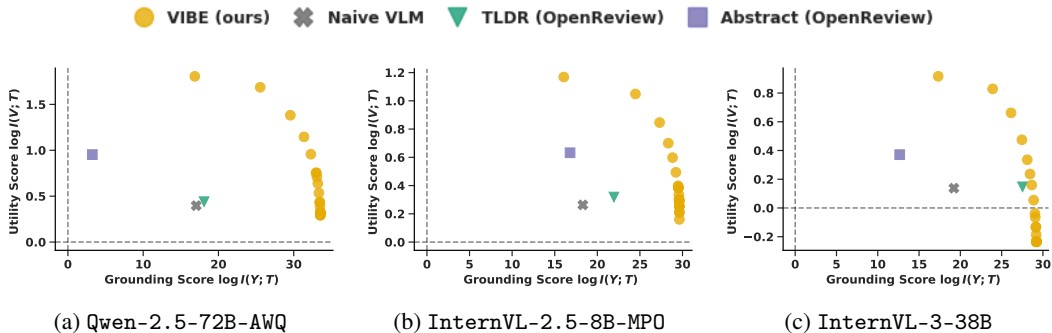

(a) `Qwen-2.5-72B-AWQ`       (b) `InternVL-2.5-8B-MPO`       (c) `InternVL-3-38B`

Figure 6: **VIBE generalizes to various VLMs.** Summaries selected by VIBE form a Pareto frontier not only across various $(\alpha, \beta)$ but also across various VLMs from different sources.

We conduct an ablation study to examine how different VLM variants affect the reliability of VIBE. Specifically, we compare QwenVL-2.5-72B-AWQ [40] (used in the main paper) with InternVL2.5-8B-MPO [47] and InternVL3-38B [48]. These models vary in size, architecture, and pretraining sources. We apply VIBE to the same set of summaries to compute their utility and grounding scores. Results in Figure 6 show consistent VIBE score trends across models, indicating that smaller or alternate VLMs yield similar evaluation signals while reducing inference cost and potential bias. Notably, various VLMs give similar ranges of both utility and grounding scores. The consistency across models further supports the generalizability of VIBE as a training-free framework.

## C    `LearningPaper24` **Dataset Curation**

We source the dataset from the public PaperCopilot PaperLists repository, which indexes all accepted papers at ICLR 2024 and NEURIPS 2024. To ensure data quality and completeness, we apply the

following filtering criteria: (1) the paper must have a valid OpenReview ID to guarantee access to full metadata and public reviews; (2) the associated SlidesLive presentation video must be accessible and playable, verified via a headless browser; (3) a TL;DR summary must be present on OpenReview; and (4) a primary area must be specified. After applying these filters, we retain 2,287 entries from an initial set of 2,556.

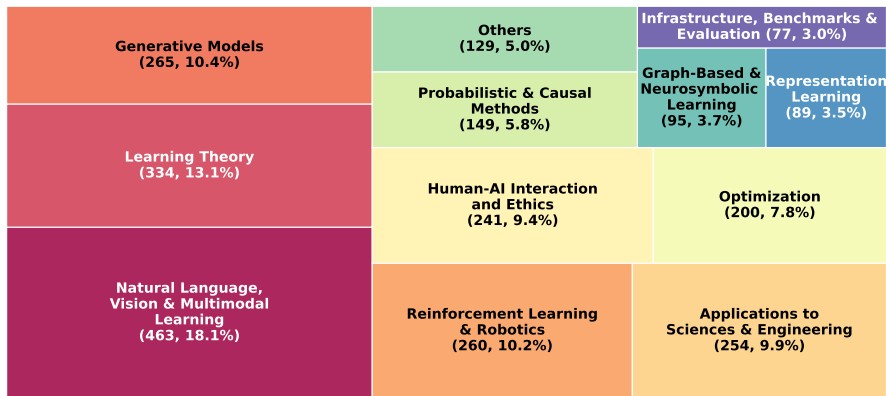

Figure 7: Distribution of papers in `LearningPaper24` by consolidated primary area.

OpenReview's primary areas are often too specific or overlapping, so we remap them into 12 broader and semantically coherent categories. For example, "self-supervised learning" and "representation learning for vision" become *Representation Learning*, while "probabilistic methods" and "causal inference" are grouped as *Probabilistic & Causal Methods*. The final taxonomy is shown in Figure 7. The final dataset includes the following required fields: OpenReview ID, SlidesLive talk ID and URL, TL;DR, abstract, and primary area.

## D   VIBE Masking and Evaluation

We now report how we select data to calculate the VIBE score in Figure 3. We further select 10 stimuli for our user study.

**LearningPaper24**   We randomly sample 80 papers each from ICLR 2024 and NeurIPS 2024 following the curation process in Appendix C. For each presentation, we generate 5 VLM responses and 1 CoT response. We compute the tf-idf score separately for each of the six response corpora. We discard words and phrases (n-grams with range 1–3) appearing in over 10% of responses and retain only those with tf-idf scores above 0.0025. We also leverage the keywords of papers provided by the authors on OpenReview for masking. We uniformly sample 20 frames from each video as VLM input, with or without random cropping applied.

**SUTD-TrafficQA**   We randomly sample 100 video clips from the dataset. For each video clip, we generate 5 VLM responses and 1 CoT response. All responses, including CoT, form the corpus for tf-idf computation. We discard words and phrases (n-grams with range 1–3) appearing in over 30% of responses and retain only those with tf-idf scores above 0.01. We uniformly select 20 frames from each video as the input of VLM with or without random cropping. We uniformly sample 20 frames from each video as VLM input, with or without random cropping applied.

**LongVideoBench**   We sample 150 video clips, each 30 to 500 seconds long, from categories `["E2O", "E2E", "O3O", "S2A", "S2E", "S2O", "SOS"]`. For each clip, we generate 5 VLM responses and 1 CoT response. All responses form the tf-idf corpus. We discard words and phrases (n-grams with range 1–3) appearing in more than 50% of responses and keep only those with tf-idf scores above 0.006. We uniformly select 20 frames from each video as the input of VLM with or without random cropping. We use the LongVideoBench package to obtain 32 frames, which is also uniformly sampling the video, from each video as VLM input, with or without random cropping applied.

# E   User Study Instruction and Interfaces

All participants provide informed consent before participation and are compensated at a rate consistent with Prolific guidelines and institutional standards. The user interface is presented in Figure 8. Below are the instructions provided to participants for each dataset. Text in parentheses indicates variations between video and text conditions.

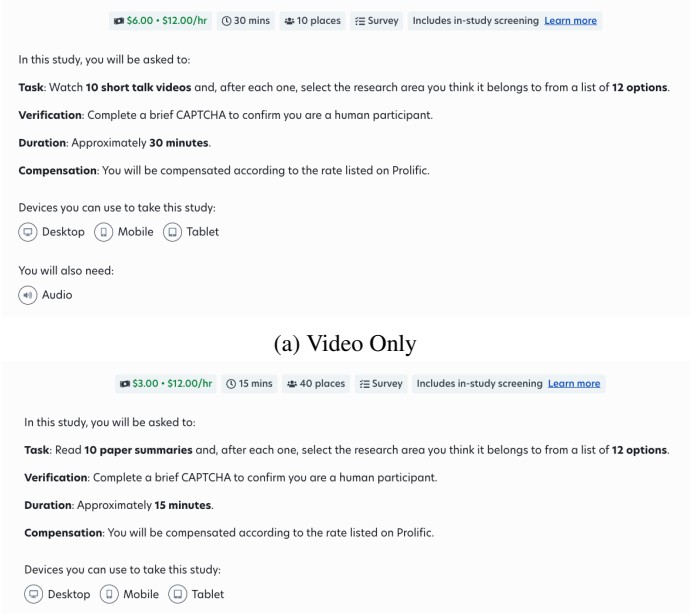

(a) Video Only

(b) Text-based (Naive, Max-U, Max-G, CoT)

Figure 8: Prolific recruitment interfaces for `LearningPaper24` dataset showing video condition (top) and text conditions (bottom). Similar interfaces are used for the other two datasets.

---

**Instructions for `LongVideoBench`**

In this study, you will (watch/read) **10 (short videos/short summaries)** featuring a variety of everyday content. Topics may include:

- Cooking
- STEM education
- Art and creativity
- Daily vlogs and lifestyle scenes

After each (video/summary), you will answer **one multiple-choice question** to assess your recall and understanding of the content.
Please **avoid leaving the page idle**, as we are also measuring your response time.

---

**Instructions for `SUTD-TrafficQA`**

In this study, you'll (watch/read) **10 (short videos/sets of descriptions)** of real traffic videos, which may involve an accident.

After each (video/description), you'll answer **4 multiple-choice questions**. These questions may involve:

- Understanding what happened in the video
- Considering what might have happened under different conditions
- Reflecting on your own interpretation or reasoning

Please **avoid leaving the page idle**, as we are also measuring your response time.

> **Instructions for `LearningPaper24`**
>
> In this study, you will (watch/read) **10 (short research talk videos/paper summaries)**.
> For each (video/summary), select the **research area** you think it belongs to from:
>
> - (A) Learning Theory
> - (B) Representation Learning
> - (C) Generative Models
> - (D) Optimization
> - (E) Probabilistic & Causal Methods
> - (F) Reinforcement Learning & Robotics
> - (G) Graph-Based & Neurosymbolic Learning
> - (H) Natural Language, Vision & Multimodal Learning
> - (I) Human-AI Interaction and Ethics (Privacy, Fairness & Safety)
> - (J) Applications to Sciences & Engineering
> - (K) Infrastructure, Benchmarks & Evaluation
> - (L) Others
>
> Please **avoid leaving the page idle**, as we are also measuring your response time.

## F  Participant Recruitment and Demographics

Table 4 summarizes demographic information for participants recruited across the three datasets.

| Dataset | # Participants | Gender Distribution | Age (years) |
|---------|----------------|---------------------|-------------|
| LearningPaper24 | 92 | 70.65% male, 28.26% female, 1.09% non-binary | $32.86 \pm 8.03$ |
| SUTD-TrafficQA | 82 | 59.76% male, 40.24% female | $40.67 \pm 12.69$ |
| LongVideoBench | 69 | 57.97% male, 40.58% female, 1.45% non-binary | $40.22 \pm 12.40$ |
| Total | 243 | 63.37% male, 35.80% female, 0.82% non-binary | $37.59 \pm 11.06$ |

Table 4: Demographic breakdown of participants by dataset.

Participants were assigned to IV conditions as shown in Table 5. In addition to the four primary conditions—Max-U, Max-G, Naive, and Video Only (see Section 5.2)—each dataset also includes a group evaluating Chain-of-Thought (CoT) responses. CoT responses are excluded from the scatter plots in the main text due to their fundamentally different generation process: unlike other conditions, CoT has access to answer options of the task, making direct comparisons unfair and potentially misleading. However, we include CoT responses in the correlation analyses, as their grounding and utility scores remain valid for assessing alignment with human performance.

| Dataset | Max-U | Max-G | Naive | Video Only | CoT |
|---------|-------|-------|-------|------------|-----|
| LearningPaper24 | 19 | 16 | 19 | 20 | 18 |
| SUTD-TrafficQA | 14 | 17 | 15 | 16 | 20 |
| LongVideoBench | 15 | 14 | 15 | 10 | 15 |

Table 5: Number of participants per IV condition across datasets.

## G  Additional User Study Plots

Figure 9 presents the full set of scatter plots and Spearman correlation results for all datasets. The left panels show scatter plots of accuracy versus inverse response time, while the right panels display correlations between accuracy and utility score, response time and utility score, accuracy and grounding score, response time and grounding score, accuracy and word count, and response time per word and word count.

## H Computation Resource

All experiments were conducted on four NVIDIA RTX 6000 Ada GPUs (48GB VRAM) using the vLLM backend. The system was equipped with an Intel(R) Xeon(R) Gold 6346 CPU @ 3.10GHz, featuring 64 cores (x86_64, 64-bit). This setup handled all models—Qwen-2.5-72B-AWQ, InternVL-2.5-8B-MPO, and InternVL-3-38B—efficiently for both generation and evaluation.

## I Derivation of Mutual Information Approximation

We derive the case where the approximation holds under the stated assumption, starting from the right-hand side of Eq. 3 and Eq. 4. Since both approximations follow nearly identical steps, we present the derivation of grounding score as an example:

$$\log \frac{\mathbf{P}(T \mid V, T_{\text{masked}})}{\mathbf{P}(T \mid T_{\text{masked}})}.$$

Since $T_{\text{masked}}$ is a masked sentence with all keywords removed, we assume it contains no information and is independent from any other random variables. Thus, we can rewrite the expression as:

$$
\begin{aligned}
\log \frac{\mathbf{P}(T \mid V, T_{\text{masked}})}{\mathbf{P}(T \mid T_{\text{masked}})} &= \log \frac{\mathbf{P}(T, V, T_{\text{masked}})/\mathbf{P}(V, T_{\text{masked}})}{\mathbf{P}(T, T_{\text{masked}})/\mathbf{P}(T_{\text{masked}})} \\
&= \log \frac{\mathbf{P}(T, V) \cdot \mathbf{P}(T_{\text{masked}})/(\mathbf{P}(V) \cdot \mathbf{P}(T_{\text{masked}}))}{\mathbf{P}(T) \cdot \mathbf{P}(T_{\text{masked}})/\mathbf{P}(T_{\text{masked}})} \\
&= \log \frac{\mathbf{P}(T, V)/\mathbf{P}(V)}{\mathbf{P}(T)} \\
&= \log \frac{\mathbf{P}(T \mid V)}{\mathbf{P}(T)}.
\end{aligned}
$$

All steps follow from Bayes' theorem and the assumption of independence.

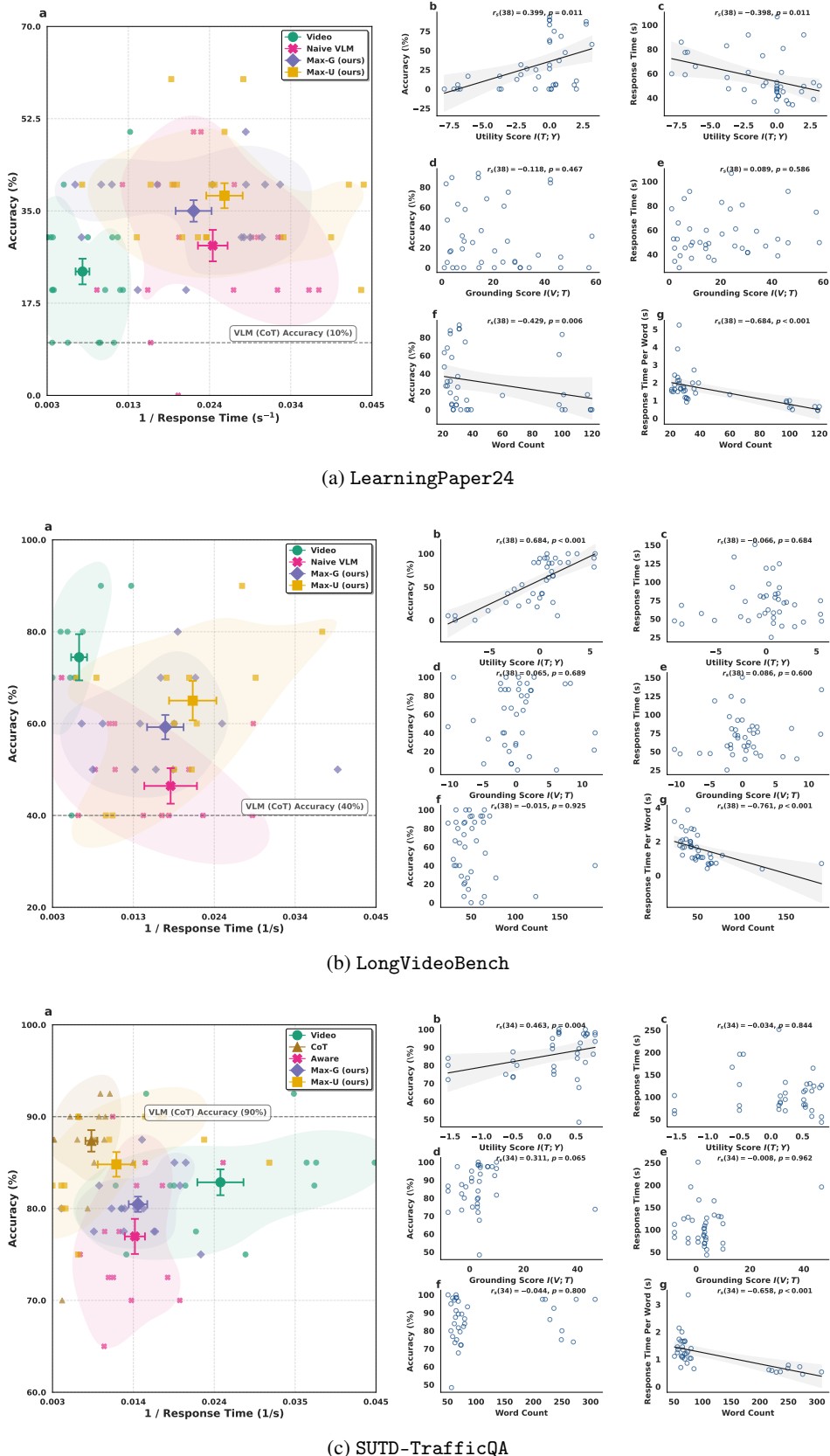

(a) `LearningPaper24`

(b) `LongVideoBench`

(c) `SUTD-TrafficQA`

Figure 9: Scatter plots and Spearman correlations across all datasets. Trendlines shown for $p < 0.05$.

