# OpenReview forum: "VIBE: Annotation-Free Video-to-Text Information Bottleneck Evaluation for TL;DR"
_NeurIPS.cc/2025/Conference — NeurIPS 2025 poster_

### Official Review · Reviewer_P4an · 2025-06-23

**Clarity:** 4
**Significance:** 3
**Originality:** 3
**Rating:** 5
**Confidence:** 3

**Summary:**

The paper presents a new method to select the best video summary for downstream tasks using video-to-text information bottleneck, without any human annotation. To achieve it, the authors propose to calculate two scores - a grounding score, which measures alignment between a summary and a video, and an utility score, which demonstrates how good summary is for the further task. There conducted experiments across three benchmarks showing that their method significantly improves people speed and accuracy on video-related tasks in most cases.

**Questions:**

1. I like the baselines you propose (video watching and random summary), but why don't you include a baseline where a meta LLM (or VLM) selects the best summary given the downstream task? Alternatively, what about a baseline where a human chooses the best summary? Also, it would be interesting to compare the performance of such a meta LLM against human annotator.
2. Could you clarify how you obtained the five different summaries used in the experiment? Did you use multiple VLMs, a single VLM with varied prompts, or some specific API? If so, please list used ones.
3. In Figure 4, there is a mention of "VLM (CoT) Accuracy (10%)", but this term is not defined in the main text. Please correct this.
4. I have a concern regarding your NeurIPS Paper Checklist response ("No") to question #8. The guidelines state: "The paper should provide the amount of compute required for each of the individual experimental runs as well as estimate the total compute." Since your experiments involve generating five summaries (using some VLMs or APIs) and calculating utility and grounding scores via VLM inference, this clearly requires compute. Please report the compute usage in the paper and revise your answer to question #8, or clarify if my understanding is incorrect.

**Ethical Concerns:**

["NO or VERY MINOR ethics concerns only"]

**Final Justification:**

I consider this paper to be a strong piece of work. All my concerns have been addressed. Therefore, I maintain score 5.

**Limitations:**

yes

**Quality:**

3

**Strengths And Weaknesses:**

**Strengths:**
1. Well-written and easy-to-follow text with excellent illustrations.
2. Introduces a novel problem and provides an elegant solution that does not require human annotation.
3. Well designed experiments involving humans, demonstrating the practical utility of the proposed method in downstream tasks.
4. Two proposed metrics enable the method to be effective both when the downstream task is known and when it is not.


**Weaknesses:**
1. Lack of an easy baseline.
2. Requires multiple inferences of the VLM for each summary.

---

> ### Author Rebuttal · Authors · 2025-07-29
>
> We thank the reviewer for thoughtful comments and detailed technical questions. Below, we respond to each point.
>
> ---
> **Weakness 1 & Question 1: Suggested additional baselines**
> Due to rebuttal time constraints, we were unable to conduct a full-scale human preference study. Additionally, such studies are often subject to annotator disagreement and noise, and more importantly, they contradict our core motivation: building an annotation-free, scalable video-caption evaluation framework that avoids the cost and subjectivity of manual labeling.
>
> Per reviewer suggestion, we experimented with GPT-4o (via OpenAI API) as a meta-evaluator on 150 examples from LongVideoBench, used in Fig. 3. For each video, GPT-4o was asked to rank five generated captions in terms of grounding and utility (the relevance to video content and questions).
>
> However, the results revealed significant issues:
> - GPT-4o indicated that **32%** of the video clips contained captions completely irrelevant to the video content, making grounding-based ranking unreliable. This issue happens even if the model is not asked to acknowledge the "unrankability".
> - Similarly,  GPT-4o flagged **92%** of captions as irrelevant to the associated task questions, making utility-based ranking equally unstable.
>
> This suggests that even strong VLMs like GPT-4o fail to reliably evaluate captions on LongVideoBench—likely due to known challenges in long-video understanding. Although humans can often infer correct answers from these captions, current VLMs cannot. Due to this inconsistency, we decided not to include GPT-4o as a baseline in our experiments.
> Interestingly, despite the poor performance of VLMs in this task, they can still be used in VIBE to evaluate captions since we do not require complicated visual reasoning, but only ask the VLM to "guess" certain masked tokens and measure their confidence in next-token probability.
>
> For the subset of examples where ranking was possible, the Mean Absolute Error (MAE) of the ranking was:
> - **Grounding MAE:** 1.59
> - **Utility MAE:** 1.92
>
> These numbers reflect disagreement with VIBE as there are only 5 captions (ranking 1~5).
> To sum up, the experiments do not justify including GPT-4o as a reliable oracle.
>
> ---
> **Weakness 2: Multiple inference limitation**
> We acknowledge this limitation, as noted in the paper. However, we wish to clarify that VIBE operates at the token level—each evaluation is a single forward pass for one token, rather than a full sequence generation. Since sentence generation involves many such token-level steps, the relative cost of VIBE is comparable and not prohibitive in practice.
> We will add a brief clarification to the limitations section to make this point more explicit.
>
> ---
> **Question 2: Implementation details**
> In all experiments (including the ablation in Appendix B), we generated **five summaries per model** using **five temperature values**: $0$, $0.5$, $1$, $1.25$, and $1.5$.
> For each run, we held the prompt, model, and vLLM backend constant. The primary model used in the main paper was **Qwen-2.5-72B-AWQ**. For ablation studies, we also evaluated:
> - **InternVL-2.5-8B-MPO**
> - **InternVL-3-38B**
> These results are reported in Appendix B.
>
> ---
> **Question 3: Clarification on “VLM (CoT) Accuracy (xx%)”**
> Thank you for pointing this out. “VLM (CoT) Accuracy (xx%)” refers to a baseline where we prompt the same VLM—Qwen-2.5-72B-AWQ—to answer the 10 task questions from the user study using only the video input and **chain-of-thought (CoT)** prompting. The percentage reflects its accuracy on those questions.
> We will clarify this definition in the main text in future revisions.
>
> ---
> **Question 4: Hardware details for reproducibility**
>
> Thank you for the reminder. We have updated the paper checklist to reflect this and added the following system information:
>
> All experiments were conducted on **four NVIDIA RTX 6000 Ada GPUs** (48GB VRAM) using the **vLLM backend**.
> The system was equipped with an **Intel(R) Xeon(R) Gold 6346 CPU @ 3.10GHz**, featuring **64 cores (x86_64, 64-bit)**.
> This setup handled all model variants—Qwen-2.5-72B-AWQ, InternVL-2.5-8B-MPO, and InternVL-3-38B—efficiently for both caption generation and VIBE evaluation.

---

> > ### Comment · Reviewer_P4an · 2025-08-03
> >
> > Thank you for your clarifications. The explanation regarding the proposed baselines is thorough, and I agree with your decision not to include them. I will keep my score unchanged—this is a high-quality paper.

---

### Official Review · Reviewer_2iXL · 2025-07-02

**Clarity:** 4
**Significance:** 2
**Originality:** 3
**Rating:** 4
**Confidence:** 4

**Summary:**

This paper proposes an **annotation-free** method for selecting video summaries generated by vision-language models (VLMs). Inspired by the Information Bottleneck framework, the authors introduce two evaluation metrics: a **grounding score**, which measures the alignment between the summary and the video content, and a **utility score**, which reflects the summary’s usefulness for downstream tasks. To make the computation tractable, the method replaces the intractable joint and marginal probability distributions with **masked-content-conditional probability distributions**, enabling the use of VLMs’ next-token prediction capabilities. The paper is well-written, and both the motivation and method are easy to understand. Human studies demonstrate that the proposed method helps people to improve task accuracy and reduce response time compared to both randomly selected summaries and human-written summaries in benchmark video summarization datasets.

**Questions:**

1. Could you provide a comparison between summaries selected by traditional metrics and those selected by your proposed method? Specifically, it would be helpful to include statistics on the agreement or disagreement between the rankings, as well as the values of your selected summaries under these traditional metrics. This would offer more insight into how your method differs from or complements existing approaches.
2. It seems that, the grounding score is normalized to the range [0, 1], while the utility score is not in Figure 3 of main paper. If these two scores are combined via a weighted sum to perform trade-offs, would it be better to normalize both for consistency and interpretability?
3. Have you considered or analyzed the lengths of the selected summaries? As shown in Tables 2 and 3 of Appendix A, the summaries selected by the proposed method appear to be longer than TL;DR but shorter than abstract, which suggests that length may be a critical factor influencing both task accuracy and response time.

**Ethical Concerns:**

["NO or VERY MINOR ethics concerns only"]

**Final Justification:**

The rebuttal clarified my main questions. I have no remaining concerns and will maintain my original score.

**Limitations:**

Yes.

**Paper Formatting Concerns:**

None.

**Quality:**

3

**Strengths And Weaknesses:**

**Strengths**
1. The method eliminates the need for costly manual annotations as references.
2. It leverages powerful pre-trained vision-language models (VLMs) as evaluators.
3. The proposed method is computationally efficient and easy to implement.

**Weaknesses**
1. Although the paper mentions **traditional metrics** such as ROUGE, BLEU, and CIDEr, it does not provide a thorough analysis or comparison between these established metrics and the proposed grounding/utility scores.
2. The proposed metrics emphasize informativeness but may overlook the **correctness** of the summaries.

---

> ### Author Rebuttal · Authors · 2025-07-29
>
> We thank the reviewer for the thoughtful feedback and detailed technical questions. Below, we respond to each point.
>
> ---
> **Weakness 1: Comparison with traditional caption evaluation metrics**
> Traditional metrics such as ROUGE, BLEU, and CIDEr require gold-standard captions, which are unavailable and prohibitively expensive to produce—particularly for the datasets we use. More importantly, these metrics evaluate sentence-to-sentence (caption-to-caption) similarity, whereas VIBE evaluates caption-to-video alignment in a multimodal setting. Hence, VIBE focuses on a fundamentally different problem.
>
> Gold-standard captions also introduce annotator bias and often lack the conciseness and utility needed for downstream tasks. Therefore, previous works need 3~5 human annotators per video. We believe comparing VIBE with caption-based metrics is not a meaningful apples-to-apples comparison.
>
> ---
> **Question 1: Why no direct comparison to BLEU or CIDEr?**
> We cannot provide a direct comparison to BLEU or CIDEr because our datasets do not include gold-standard captions, which these metrics require. This is precisely why we propose VIBE: to support evaluation in the absence of reference annotations.
>
> Furthermore, even when available, high BLEU scores between candidate summaries do not guarantee correctness—if all captions are inaccurate, their mutual similarity is irrelevant. Our method instead evaluates how well a generated summary aligns with the video content, a dimension traditional caption-to-caption metrics fail to capture.
>
> As suggested by **Reviewer P4an**, we have added an additional baseline using LLM-as-a-judge for caption selection.
> Specifically, we experimented with GPT-4o (via OpenAI API) as a meta-evaluator on 150 examples from LongVideoBench, used in Fig. 3. For each video, GPT-4o was asked to rank five generated captions in terms of grounding and utility (the relevance to video content and questions).
>
> However, the results revealed significant issues:
> - GPT-4o indicated that **32%** of the video clips contained captions completely irrelevant to the video content, making grounding-based ranking unreliable. This issues happens even if the model is not asked to acknowledge the "unrankability".
> - Similarly,  GPT-4o flagged **92%** of captions as irrelevant to the associated task questions, making utility-based ranking equally unstable.
>
> This suggests that even strong VLMs like GPT-4o fail to reliably evaluate captions on LongVideoBench—likely due to known challenges in long-video understanding. Although humans can often infer correct answers from these captions, current VLMs cannot. Due to this inconsistency, we decided not to include GPT-4o as a baseline in our experiments.
> Interestingly, despite the poor performance of VLMs in this task, they can still be used in VIBE to evaluate captions since we do not require complicated visual reasoning, but only asking the VLM to "guess" certain masked tokens and measure their confidence in next-token probability.
>
> For the subset of examples where ranking was possible, the Mean Absolute Error (MAE) of the ranking was:
> - **Grounding MAE:** 1.59
> - **Utility MAE:** 1.92
>
> These numbers reflect disagreement with VIBE as there are only 5 captions (ranking 1~5).
> To sum up, the experiments do not justify including GPT-4o as a reliable oracle.
>
> ---
> **Weakness 2: On summary correctness**
> Regarding task correctness, we demonstrated in Fig. 4(a) and (b) that VIBE scores correlate with human task performance, as reflected by accuracy and utility ratings.
>
> Our grounding score assesses factual correctness by measuring how token probabilities shift when the video is removed. Tokens whose probabilities remain unchanged are likely ungrounded (Eq. 3), suggesting the model generated them without relying on visual evidence. As a result, hallucinated content typically receives grounding scores near zero. Similarly, when a caption is factually incorrect—either irrelevant or contradictory—inputting the video tends to suppress the probability of those tokens, leading to zero or negative scores. This mechanism allows our method to identify ungrounded or inaccurate content.
>
> ---
> **Question 2: Clarification on score range and normalization**
> We believe the reviewer is referring to the utility score, as it has the closest range to [0, 1]. We also acknowledge that Fig. 3 had the axes flipped: the X-axis should be “Grounding Score $\log I(V; T)$” and the Y-axis should be “Utility Score $\log I(Y; T)$”. This has been corrected.
>
> As noted in Line 132, pointwise mutual information (PMI) can range from \([−∞, ∞]\). While the scores may appear bounded in Fig. 3, this is not mathematically the case—both grounding and utility scores are unbounded in theory.
>
> Regarding score behavior: the grounding score (Eq. 3) is computed at the token level, while the utility score considers the joint probability over multiple masked tokens. This means utility scores tend to be higher when more relevant keywords appear. We considered normalizing by the number of masked tokens, but doing so would break consistency with mutual information theory and would make $\alpha$ a function of the number of masked tokens.
>
> ---
> **Question 3: Influence of summary length on response time and accuracy**
> We computed the mean and standard deviation of word count for each method across datasets:
> | Dataset         | Naive VLM            | Max-U                | Max-G                |
> |-----------------|----------------------|-----------------------|----------------------|
> | LearningPaper   | $27.90 \pm 4.79$     | $30.20 \pm 11.38$     | $29.90 \pm 5.36$     |
> | LongVideoBench  | $48.60 \pm 7.18$     | $38.20 \pm 9.05$      | $42.40 \pm 10.27$    |
> | SUTD-TrafficQA  | $65.60 \pm 10.02$    | $67.60 \pm 7.09$      | $68.00 \pm 7.60$     |
>
> As shown, the summary lengths are relatively consistent across methods within each dataset. This is expected, as all methods select from the same candidate pool with comparable length distributions. Therefore, differences in task accuracy and response time found in Table 1 are unlikely to be explained by length alone.
>
> As shown in Appendix Fig. 9(f), we conducted a correlation analysis between summary length and task accuracy across all three datasets:
> * **LongVideoBench**: No significant correlation
> * **SUTD-TrafficQA**: No significant correlation
> * **LearningPaper24**: A weak positive correlation was observed (Spearman $r\_s(38) = 0.429,\ p = 0.006$)
>
> These results suggest that while summary length may have some influence, particularly in the LearningPaper dataset, it is not a consistent or dominant factor. Instead, accuracy appears to be better predicted by the utility score we proposed. We conjecture that **task accuracy depends more on the presence of informative content (e.g., task-relevant keywords)** than on overall word count.
>
> Regarding response time, we did not observe a significant correlation with summary length in any dataset. This may be due to the narrow length distribution across conditions, as all summaries were drawn from the same candidate pool. While it is intuitive that longer summaries require higher response time, we consider this effect an inherent property of human information processing and hence not a focus of our analysis.

---

> > ### Comment · Reviewer_2iXL · 2025-08-02
> >
> > Thank you for the detailed response. Your explanations have addressed my concerns. I will maintain my original score.

---

### Official Review · Reviewer_SJcs · 2025-07-03

**Clarity:** 3
**Significance:** 3
**Originality:** 3
**Rating:** 5
**Confidence:** 3

**Summary:**

This paper presents an annotation-free framework for evaluating video-to-text summaries generated by VLMs. Grounding and utility score are introduced to evaluate the VLM outputs. Grounding score measures the alignment of summary semantics with visual signal, while utility score measures the task relevance of the generated summary. User studies on several benchmarks demonstrate the method's effectiveness.

**Questions:**

Mentioned in the Weaknesses section.

**Ethical Concerns:**

["NO or VERY MINOR ethics concerns only"]

**Final Justification:**

The rebuttal has addressed my concerns, and many of those of other reviewers. Therefore, I have decided to maintain my positive score.

**Limitations:**

See my cons.

**Paper Formatting Concerns:**

No.

**Quality:**

3

**Strengths And Weaknesses:**

Strengths:

1. A novel video caption evaluation method. The idea of decoupling the evaluation score to grounding and utility makes sense to me.
2. The manuscript clearly communicates its key contributions and ideas.
3. The user studies are comprehensive, indicating the method's effectiveness.

Weaknesses:

1. It seems that the grounding score can not prevent the summary from the hallucination problem without additional visual cues, as discussed in previous works [1][2]. The grounding score may be high even when there are factual errors between the summary and the video sequence (just with high semantic similarity).
2. In lines 282 thru 284 mention "the brevity of the clips and the visual nature of fine-grained actions reduce the benefit of text-based summaries." Does this mean the proposed method may suffer from object-centric bias?

[1]Li, Chaoyu, Eun Woo Im, and Pooyan Fazli. "Vidhalluc: Evaluating temporal hallucinations in multimodal large language models for video understanding." *Proceedings of the Computer Vision and Pattern Recognition Conference*. 2025.

[2]Yin, Shukang, et al. "Woodpecker: Hallucination correction for multimodal large language models." arXiv preprint arXiv 2310.16045, 2023.

---

> ### Author Rebuttal · Authors · 2025-07-29
>
> We thank the reviewer for the insightful comments and for pointing to relevant related work. Below, we address the specific concerns raised.
>
> ---
> **Weakness 1: On grounding score and hallucination**
> The grounding score measures the logarithm of the "lift" in next-token probability with and without the video, given a masked summary (Eq. 3).
> Roughly speaking, hallucination can arise from two sources:
>
> 1. **Internal model bias** — originating from pretraining or fine-tuning;
> 2. **External input** — due to misleading or misinterpreted content from the video or summary.
>
> If hallucination stems from **(1)**, the model’s predictions likely remain unchanged regardless of the video, and the grounding score becomes ($\log 1 = 0$), indicating the summary is ungrounded.
>
> If it arises from **(2)**—due to misleading content or VLM misinterpretation or mis-attribution of key frames—the summary is still grounded, because the video influences generation. The masked summary, with removal of meaningful keywords, should carry no new information and thus should not contribute to hallucination
>
> The works referenced by the reviewer ([1], [2]) primarily examine the fundamental limitations of VLMs,  which can be categorized into case **(1)**. Since the grounding score also considers the accuracy (probability of answering the correct option) without video, this term acts as a normalization term to mitigate the internal model bias.
>
> ---
> **Weakness 2: Object-centric bias and framing of short clips**
> We agree that object-centric bias is a known limitation of some VLMs, but this issue stems from model pretraining, not from our proposed evaluation method. Recent VLMs are typically trained to mitigate this bias, and we do not observe it to significantly impact our results.
>
> As noted in Lines 282–284, traffic events in SUTD-TrafficQA unfold over just a few seconds. In such settings, directly viewing the clip is often more efficient and informative than reading textual summaries—even when the sampled frames inputted by VLM capture the full event and the generated summary is accurate. This emphasizes VIBE’s value in scenarios where processing full videos is costly or time-constrained.

---

> > ### Comment · Reviewer_SJcs · 2025-08-05
> >
> > The rebuttal addressed my concerns, so I decide to maintain my positive score.

---

### Official Review · Reviewer_iMoZ · 2025-07-07

**Clarity:** 3
**Significance:** 2
**Originality:** 2
**Rating:** 4
**Confidence:** 3

**Summary:**

Considering the existing video caption evaluation metrics either require human annotators to write gold-standard captions or fail to measure how well captions support the downstream tasks. The paper introduces VIBE (Video-to-text Information Bottleneck Evaluation), an annotation-free method for evaluating and selecting video-to-text summaries to support human decision-making. VIBE uses grounding and utility scores, based on the information bottleneck principle to evaluate video-to-text summaries. The grounding score measures how well the summary aligns with the visual content, while the utility score assesses how informative the summary is for a downstream task. By conducting between-subjects user studies with 243 participants across three datasets, the results demonstrate that summaries selected by VIBE significantly improve human task accuracy.

**Questions:**

As provided in Section Strengths And Weaknesses, W1-W6.

**Ethical Concerns:**

["NO or VERY MINOR ethics concerns only"]

**Final Justification:**

The authors' rebuttal was constructive and has satisfactorily addressed most of my initial concerns. I appreciate the authors' detailed responses.

**Limitations:**

Yes

**Quality:**

2

**Strengths And Weaknesses:**

S1: VIBE introduces an annotation-free method for evaluating video-to-text summaries, using grounding and utility metrics, which innovatively introduced Mutual Information and Information Bottleneck to score VLM-generated summaries.
S2: The authors conduct extensive user studies across three diverse datasets involving 243 participants. The results demonstrate that summaries selected by VIBE significantly improve human task accuracy by up to 61.23% and reduce response time by up to 75.77% compared to naive VLM summaries or watching the full video.
W1: Inadequate coverage of VLMs in related work. The study heavily relies on VLMs, yet the related work section lacks a comprehensive review of the state-of-the-art in VLM research. The author should expand the related work to include a detailed discussion of VLM development and their application in caption.
W2: The technical novelty presented in this paper is somewhat limited. The proposed VIBE heavily draws from existing literature. The novelty of this paper can be illustrated better. Specifically, the challenges authors want to address should be highlighted. The techniques to illustrate challenges should be more detailed.
W3: The description of the approximation using pointwise mutual information is insufficient.: VIBE introduces the use of masked information to approximate the grounding score and utility score. The author should provide a deeper explanation and provide a detailed derivation of the formula to justify the approximation, which will give the reader a better understanding of the reasons for this operation.
W4: As mentioned in abstract, the author argue that the VLMs would have generated verbose caption. The authors should incorporate additional quantifiable metrics, such as average output token lengths to provide a more comprehensive evaluation of redundancy reduction. This would better validate VIBE’s impact on optimize the length of the output.
W5: More explanations on experimental results. The authors should provide potential reasons, e.g. the similar performance of Video-Only and Max-G on SUTD-TrafficQA. This would enhance the interpretability of the findings.
W6: The authors should provide a clearer explanation of the applicable scenarios for their work. While VIBE is designed to support human decision-making, the discussion of limitations in traditional caption evaluation metrics may be confusing. To better illustrate the problem addressed by this work, the authors could offer an alternative example that demonstrates the need for an approach like VIBE instead of the examples of evaluation metrics.

---

> ### Author Rebuttal · Authors · 2025-07-29
>
> We thank the reviewer for the thoughtful and detailed feedback. Below, we respond to each concern individually.
>
> ---
> **Weakness 1: Missing coverage of VLMs in Related Work**
> We have added a subsection in the related work to address recent advancements in video-language models (VLMs). This new section surveys key developments in VLM architecture, training, and evaluation benchmarks--including models not directly used in the paper [1, 2, 3].
>
> We also discuss how these models have been applied in related tasks such as video captioning [4], video understanding, and multimodal reasoning [5], to contextualize our approach within the broader landscape of VLM research. In the final version, we will further expand this subsection to include additional relevant works beyond the citations listed here.
>
> [1] Nguyen, Thong, et al. "Video-language understanding: A survey from model architecture, model training, and data perspectives." arXiv preprint arXiv:2406.05615 (2024).
>
> [2] Tang, Yunlong, et al. "Video understanding with large language models: A survey." IEEE Transactions on Circuits and Systems for Video Technology (2025).
>
> [3] Li, Zongxia, et al. "A survey of state of the art large vision language models: Alignment, benchmark, evaluations and challenges." arXiv preprint arXiv:2501.02189 (2025).
>
> [4] Abdar, Moloud, et al. "A review of deep learning for video captioning." IEEE Transactions on Pattern Analysis and Machine Intelligence (2024).
>
> [5] Li, Yunxin, et al. "Perception, reason, think, and plan: A survey on large multimodal reasoning models." arXiv preprint arXiv:2505.04921 (2025).
>
> ---
> **Weakness 2: Clarification on the challenge and novelty**
> We appreciate the opportunity to clarify this. Our key challenge lies in evaluating video-language alignment in unannotated, real-world videos without human-annotated captions—a setting poorly addressed by existing metrics. Unlike prior work that assumes access to human-annotated gold-standard text (e.g., captions or scripts), our approach enables scalable evaluation directly from raw video and model-generated text.
>
> This setting introduces both conceptual and technical innovations. Conceptually, it redefines the evaluation of "video captioning" without relying on human annotations. Technically, it grounds evaluation in mutual information approximations between modalities, as detailed in Section 4.
>
> In contrast, prior works in the related work section applied mutual information with the token masking technique only to assess text-to-text summary quality.
>
> ---
> **Weakness 3: Mathematical derivation to justify approximation using pointwise mutual information**
>
> We derive the case where the approximation holds under the stated assumption, starting from the right-hand side of Eq. 3 and 4. Since both approximations follow nearly identical steps, we present the derivation of the grounding score as an example:
> $$
> \log \frac{\mathbf{P}(T \mid V, T_{\mathrm{masked}})}{\mathbf{P}(T \mid T_{\mathrm{masked}})}.
> $$
>
> Since $T_{\mathrm{masked}}$ is a masked sentence with all keywords removed, we assume it contains no information and is independent of any other random variables. Thus, we can rewrite the expression as:
>
> $$
> \log \frac{\mathbf{P}(T \mid V, T_{\mathrm{masked}})}{\mathbf{P}(T \mid T_{\mathrm{masked}})}
> = \log \frac{\mathbf{P}(T, V, T_{\mathrm{masked}}) / \mathbf{P}(V, T_{\mathrm{masked}})}{\mathbf{P}(T, T_{\mathrm{masked}}) / \mathbf{P}(T_{\mathrm{masked}})}
> = \log \frac{\mathbf{P}(T, V) \cdot \mathbf{P}(T_{\mathrm{masked}}) / (\mathbf{P}(V) \cdot \mathbf{P}(T_{\mathrm{masked}}))}{\mathbf{P}(T) \cdot \mathbf{P}(T_{\mathrm{masked}}) / \mathbf{P}(T_{\mathrm{masked}})}
> = \log \frac{\mathbf{P}(T, V)/\mathbf{P}(V)}{\mathbf{P}(T)}
> = \log \frac{\mathbf{P}(T \mid V)}{\mathbf{P}(T)}.
> $$
>
> All steps follow from Bayes’ theorem and the assumption of independence.
>
> ---
> **Weakness 4: Metrics on redundancy reduction**
>
> Thank you for the suggestion. Recent works [1,2] on long video understanding with VLMs often generate clip-level captions and aggregate them for downstream tasks, resulting in a lengthy and verbose summary composed of many sentences.
> While we briefly mention this in the abstract, it is not the central focus of our study. Our current experiments emphasize downstream task performance rather than length reduction.
>
> That said, we believe VIBE offers a principled approach to addressing verbosity—not by simply truncating output, which can compromise grounding and utility—but by selectively preserving the most informative content. We have noted verbosity evaluation as a promising direction in the future work section.
>
> [1] Hur, Chan, et al. "Narrating the Video: Boosting Text-Video Retrieval via Comprehensive Utilization of Frame-Level Captions." Proceedings of the Computer Vision and Pattern Recognition Conference. 2025.
>
> [2] Wang, Ziyang, et al. "Videotree: Adaptive tree-based video representation for LLM reasoning on long videos." Proceedings of the Computer Vision and Pattern Recognition Conference. 2025.
>
> ---
> **Weakness 5: Interpretation of SUTD-TrafficQA results**
> Thank you for raising this point. If the paper is accepted, we will expand the interpretation in the final version to further contextualize the results.
>
> Regarding the specific findings on SUTD-TrafficQA: As shown in Table 1, while the accuracy of Video-Only and Max-G is similar, the key differentiator is response time (Video-Only: 48.63s; Max-G: 80.97s; Max-U: 137.27s). As discussed in Lines 272–275 and 297–298, this pattern is due to the short video durations (2–10 seconds) in SUTD-TrafficQA. In such settings, participants can extract relevant information directly from the video with low working memory demand. In contrast, text summaries require parsing language and mentally reconstructing events—particularly challenging for temporally and spatially grounded traffic event questions.
>
> This result suggests a promising future direction: systematically identifying the boundary conditions under which summarization enhances human task performance versus when direct video consumption is more effective. Exploring this trade-off more systematically could inform adaptive interfaces that tailor content presentation to context. Despite this open question, the consistent utility-accuracy correlation trends across datasets (see Appendix Fig. 9 (b)) support the generalizability of our core findings. Moreover, video-to-text summarization methods offer clear value in scenarios where visual attention is constrained—such as accessibility applications, mobile multitasking, or hands-free settings—where direct video inspection may not be feasible and text can be converted to audio for easier accessibility.
>
> ---
> **Weakness 6: Elaboration on Applicable Scenarios**
> We appreciate the chance to elaborate on our evaluation setting. Our work focuses on real-world scenarios where videos often lack human-generated captions. This is typical in surveillance, driving, sports, and other domains where scalable annotation is infeasible. As such, conventional reference-based metrics (e.g., CIDEr, METEOR) are not applicable.
> The datasets we selected reflect this challenge: they contain raw video and questions but no canonical ground-truth summaries. Our approach—VIBE—avoids reliance on expensive or biased annotations by directly measuring alignment between video and model-generated summaries. We believe this setup is not only practical but essential for general-purpose video understanding evaluation going forward.

---

> > ### Author Response · Authors · 2025-08-05
> > **Follow up on our rebuttal**
> >
> > Dear Reviewer iMoZ,
> >
> > We are writing to kindly follow up on our rebuttal. If you have any thoughts or feedback, we would greatly appreciate your response or any points you would like to discuss.

---

### Note · Authors · 2025-08-11

We thank the reviewers for the constructive feedback. Despite Reviewer iMoZ not responding to our rebuttal, interactions with the other reviewers have clarified several key points. We summarize them here for the AC’s consideration.

**Alternative baseline**
Following the suggestions of reviewer 2iXL, we tested GPT-4o as an “LLM-as-a-judge” on 150 LongVideoBench examples. However, it flagged 32% of captions as completely irrelevant to the video and 92% as irrelevant to task questions, yielding unstable rankings. Even in the subset where ranking was possible, MAE vs. VIBE was 1.59 (grounding) and 1.92 (utility). These results highlight the current difficulty of using even strong VLMs as reliable evaluators for long-video settings, reinforcing the need for VIBE’s annotation-free design.

**Metric applicability**
Traditional caption-similarity metrics (BLEU, CIDEr) were not used because our datasets lack gold captions, by design. This reflects the real-world, annotation-scarce scenarios we target.

**Interpretation of SUTD-TrafficQA results**
The similar accuracy of Video-Only and Max-G is due to short clip lengths, where direct viewing is efficient. This supports the future direction of identifying boundary conditions under which summarization aids task performance.

**Other clarifications**
- Verbosity: While not a main focus, VIBE can guide selective content preservation without harming grounding/utility.
- Length effects: Summary length was consistent across methods and showed no consistent correlation with accuracy or response time; performance differences are better explained by utility scores.
- Mathematical derivation: We detailed the independence assumption enabling our PMI-based approximation for grounding and utility scores.

---

### Decision · Program_Chairs · 2025-09-17

**Decision:**

Accept (poster)

**Comment:**

This paper introduces an annotation-free framework for evaluating video-to-text summaries generated by VLMs, proposing two key metrics: grounding score and utility score. Prior to the rebuttal, the paper received mixed reviews from reviewers, the main concerns are about the absence of comparisons with traditional evaluation metrics and insufficient discussion of related inferences from VLMs. The authors have addressed the majority of these concerns in their rebuttal, providing clarifications and additional results that mitigate the initial criticisms. Based on this, the final recommendation is to accept this paper. We encourage the authors to incorporate the addressed points into the camera-ready version to further strengthen the work.